# Iron Therapy in Inflammatory Bowel Disease

**DOI:** 10.3390/nu12113478

**Published:** 2020-11-12

**Authors:** Aditi Kumar, Matthew J. Brookes

**Affiliations:** 1The Royal Wolverhampton NHS Trust, Wolverhampton WV10 0QP, UK; matthew.brookes@nhs.net; 2Research Institute in Healthcare Science (RIHS), University of Wolverhampton, Wolverhampton WV1 1LY, UK

**Keywords:** iron, inflammatory bowel disease, iron therapy, intravenous iron, oral iron, iron deficiency

## Abstract

The most common complication seen in inflammatory bowel disease (IBD) patients is iron deficiency anaemia (IDA). Symptoms such as chronic fatigue can be as debilitating to IBD patients as pathological symptoms of abdominal pain and diarrhoea. Recognising and correcting anaemia may be as important as managing IBD symptoms and improving overall quality of life. Thus, iron replacement should be commenced the moment IDA is identified. Although intravenous iron is now considered standard treatment for IBD patients in Europe, oral iron still appears to be the preferred option. Advantages of oral iron include greater availability, lower costs and ease of applicability. However, its multitude of side effects, impact on the microbiome and further exacerbating IBD activity can have consequences on patient compliance. The newer oral iron formulations show promising safety and efficacy data with a good side effect profile. Intravenous iron formulations bypass the gastrointestinal tract absorption thereby leading to less side effects. Multiple studies have shown its superiority compared to oral formulations although its risk for hypersensitivity reactions continue to lead to clinician hesitancy in prescribing this formulation. This article provides an updated review on diagnosis and management of IDA in IBD patients, discussing the newer oral and intravenous formulations.

## 1. Introduction

Inflammatory bowel disease (IBD) is recognized as a group of chronic relapsing–remitting diseases that affect the small and large bowel [1]. Its incidence and prevalence are rising globally, with higher rates reported in the western nations [1,2]. There are multiple extraintestinal manifestations associated with IBD with the most common complication presenting as anaemia [1]. Anaemia in IBD is detected in up to 70% of inpatients and 20% of outpatients [3] and is thought to affect one third of the IBD population at any one time [4]. Anaemia in IBD patients can be found both at the initial diagnosis stage and during repeated flares [5], more commonly seen in Crohn’s disease than Ulcerative Colitis [1], with females at higher risk than males [6]. Importantly, there is a greater risk of hospitalization and surgery rates in IBD with concomitant anaemia [7]. It is also demonstrated to be the most frequent comorbid condition associated with death, although this may just reflect the severity of the underlying IBD condition [8]. The most common causes for anaemia in IBD include iron deficiency, vitamin B12 deficiency and anaemia of chronic disease (ACD) (see Table 1). This review article will focus specifically on the management of iron deficiency anaemia (IDA).

### 1.1. Iron Absorption, Iron Deficiency and Iron Deficiency Anaemia

Iron is a vital element involved in several cellular functions, including but not limited to DNA repair, gas exchange, mitochondrial function and free radical production [9]. Dietary iron is found in two forms: haem, which arises from haemoglobin and myoglobin in the forms of meat, poultry and fish, and non-haem, which mainly comes from plants [10]. Haem iron is more efficiently absorbed than non-haem iron [11].

Iron absorption occurs predominantly in the duodenum and upper jejunum (Figure 1).

Absorption is facilitated by ascorbic acid, citrate and gastric acid and inhibited by dietary sources such as phytates, tannins and antacids, all commonly found in plants [12]. There are two absorption pathways, depending on its iron formulation. Non-haem iron is present in its ferric form (Fe^3+^) and must first be reduced to ferrous iron (Fe^2+^) by the enzyme duodenal cytochrome B (DcytB), which is present on the apical membranes of enterocytes [13]. Fe^2+^ can then be absorbed across the apical surface of duodenal enterocytes via the divalent metal transporter 1 (DMT1) [11]. Once inside the enterocyte, Fe^2+^ is stored as ferritin or exported across the basolateral surface to plasma through ferroportin [14]. However, before iron can be transported outside the enterocyte, Fe^2+^ must be oxidised back to Fe^3+^ via hephaestin or ceruloplasmin. Haem iron, on the other hand, can be absorbed directly into the enterocyte via the haem carrier protein (HCP1) and then released into plasma via the haem exporter FLVCR1. Once inside the enterocyte, Haem can also undergo degradation, releasing Fe^2+^ through the enzymatic reaction of haem oxygenase (HO). Fe^2+^ is then released into the plasma through ferroportin [13]. Ferroportin is the sole iron exporter and is controlled by hepcidin, a liver-derived peptide hormone [12,14]. Hepcidin binds to ferroportin and controls ferroportin concentration through promoting its endocytosis [14]. The presence of inflammation can increase hepcidin levels and impair the exportation of cellular iron into plasma by causing ferroportin degradation [15].

The human body can store approximately 3–4 g of iron with a resultant daily iron loss of 1–2 mg, which requires a small amount to be replenished from the individual’s diet [10,16,17]. Daily iron can be lost through skin desquamation and through the intestine [15], urinary tract and bile ducts [16,17]. The two main sources of excessive iron loss are either through menstrual bleeding or intestinal bleeding [10,16]. In IBD, chronic intestinal bleeding exceeds dietary iron absorption resulting in a negative iron balance, ultimately leading to iron deficiency anaemia [18].

The World Health Organisation (WHO) has recognised IDA as the most common disorder of the world with 30% of the population affected by this condition. WHO classifies anaemia as a haemoglobin (Hb) level less than 120 g/L for women and less than 130 g/L for men. In pregnancy and IDA, Hb should be less than 110 g/L [19]. Generally, low serum iron and ferritin levels are used to diagnose IDA [17]. However, ferritin is an acute phase protein and is increased in the presence of chronic inflammation [4]. The European Crohn’s and Colitis Organisation (ECCO) guidelines state ferritin levels should be less than 30 ug/mL with IBD patients in remission (in the absence of inflammation) or mild disease and less than 100 ug/mL in active disease [20]. Transferrin levels can also be increased in IDA, although it is a negative acute phase protein and is decreased in chronic inflammation. Transferrin saturations are generally low in IDA (<20%) and can be useful for diagnosis in unclear cases as it does not correlate with concomitant inflammation [17]. ACD and IDA can be difficult to distinguish and at times can be seen in combination with active IBD [17], where ferritin levels are between 30 and 100 ug/L, and transferrin saturations are less than 20% [20].

Iron deficiency cannot be excluded in the presence of a normal Hb as patients must lose a significant amount of body iron before the Hb levels fall [21]. Thus, a low mean corpuscular haemoglobin (MCH) with a normal Hb, or an increase in red cell distribution width (RDW) signifies mild iron deficiency without anaemia [21,22].

The aetiology of IDA in IBD patients is multifactorial. As stated above, IDA may be secondary to a reduced dietary intake or from chronic blood loss from inflamed mucosa [4]. Reduced iron absorption is another cause [1]. The majority of iron absorption occurs in the duodenum with a smaller proportion in the proximal ileum [23]. Consequently, iron absorption is impaired where there is duodenal inflammation. This is a result of elevated circulating hepcidin levels which inhibit enterocytes from releasing iron into circulation [1]. Reduced absorption can be seen in patients with extensive bowel resection with the residual complication of short bowel syndrome [4]. IDA can also be aggravated by medication used for IBD such as methotrexate, thiopurines, sulfasalazine and proton pump inhibitors [15,17,24]. Intestinal inflammation and extensive bowel resection can further perpetuate IDA by causing other vitamin deficiencies such as B12, folate and vitamin D [24,25].

### 1.2. Clinical Manifestations of IDA and Its Effect on Quality of Life

IDA can lead to a multitude of symptoms and can affect multiple organs. Common symptoms include fatigue, dizziness, headaches and shortness of breath particularly on exertion [26,27]. Signs to note include pallor of the nails, conjunctiva and skin [26]. Iron deficiency with or without anaemia can also lead to reduced exercise tolerability, skin dryness, hair breakage and restless leg syndrome [26,27]. There is also an increased risk of infection and alterations of thyroid hormones, catecholamines and neurotransmitters [17]. This can present with increased stress and depression in post-partum anaemia and poor cognitive and motor development outcomes in children. Among the elderly, this can be associated with increased hospitalisations and disability, greater risk for falls, impairment of executive function and dementia, all of which can lead to greater risk of death [28].

Chronic fatigue can affect virtually every aspect of daily life, resulting in significant biopsychosocial consequences [10]. It has been reported that chronic fatigue secondary to anaemia can debilitate IBD patients as much as organic symptoms of abdominal pain and diarrhoea [29]. In fact, it is thought that correcting patients’ anaemia may be as important as controlling their IBD symptoms in relation to the favourable effect on quality of life (QoL) [17]. Multiple studies have shown that treating iron deficiency successfully can improve QoL unrelated to IBD disease activity [29]. This improvement was credited to increased feelings of well-being and mood, improved physical abilities and a rise in social activities with a strong correlation in haemoglobin levels in IBD patients [10].

## 2. Management

It is recommended to treat all patients with IDA with or without symptoms by correcting the underlying cause and by replenishing adequate iron stores [14]. Treatment should be initiated as soon as the patient is found to have IDA [30]. The goal of treatment is to obtain complete normalisation of anaemia and iron stores [1]. Although iron replenishment given either through the oral or intravenous (IV) route is the mainstay treatment, other options include erythropoietin-stimulating agents and blood transfusions.

### 2.1. Oral Iron

They are the most commonly prescribed iron replacement therapy due to low cost, availability and ease of administration [31]. Unless patients are intolerant, oral iron is preferred with mild IDA (Hb > 100 g/L) in quiescent IBD [20]. For oral iron replacement to be effective, six months of treatment is recommended to replenish iron stores [14] and the Hb should increase by a minimum of 20 g/L within 4 weeks of treatment [1].

#### Types of Preparations

Due to its greater bioavailability and solubility, the ferrous form of iron (Fe^2+^) is generally found in the oral iron formulations [18,24]. Ferrous sulfate, ferrous gluconate and ferrous fumarate are the traditional formulations. For adequate absorption by enterocytes, Fe^3+^ must be reduced to Fe^2+^, catalysed by DcytB and enhanced by ascorbic acid [32]. The optimal oral iron dose is yet to be established with a recommended dose of 50–200 mg once daily [24]. ECCO recommends that the dose should not be higher than 100 mg of elemental iron/day, as only 10–20 mg/day of iron is effectively absorbed [24]. As side effects are dose-related [16,20], higher doses are not recommended particularly as there is no increase in iron absorption and efficacy [12]. Studies in iron-deficient women have further demonstrated that alternate-day single doses are more effective as iron absorption is reduced in more frequent daily dosing [33]. Moretti et al. established that oral iron doses of 60 mg or greater will result in higher fractional absorption when dosages are spaced by 48 h due to the transient increase in hepcidin for up to 24 h [34]. To improve absorption further, iron tablets should be taken on an empty stomach or in a mildly acidic medium such as 250 mg ascorbic acid or with a glass of orange juice [16].

### 2.2. Ferric Maltol (Ferracru)

Ferric maltol is a new oral iron formulation. It is formed from a stable complex of ferric (Fe^3+^) iron with maltol. Fe^3+^ iron has been demonstrated to be less toxic to the gastrointestinal (GI) tract mucosa. Although, its use is not approved in the United States, it has been offered as treatment for IDA in the European Union since 2016 [35]. Once ferric maltol has been ingested, ferric iron enters the intestinal mucosa in its complex form, thereby minimising the formation of free iron. This allows for more efficacious uptake of ferric iron into the enterocytes [35,36]. Furthermore, a lower daily dose of ferric maltol is required to achieve adequate iron uptake. Studies have thus far been encouraging, demonstrating rapid improvements in Hb with a favourable safety and tolerability profile [36]. The efficacy and tolerability of ferric maltol in patients with IBD and IDA was evaluated in a randomised double-blind placebo-controlled trial. This study showed that patients in the ferric maltol group achieved Hb normalisation within 12 weeks of treatment as opposed to 13% in the placebo arm. Constipation, abdominal pain and flatulence were the most common side effects. However, this study was not a direct comparison to other iron formulations, patients were only included if their disease was in remission or non-severe and if they had previously failed treatment with other oral ferrous formulations [35,36]. Further studies have confirmed an acceptable safety profile for ferric maltol, demonstrating good tolerability and the majority of patients had Hb normalisation within 3 months [3]. Howaldt et al. also demonstrated greater improvements in short form (SF-36) and mental component summary (MCS) scores with ferric maltol over IV iron, suggesting a greater improvement in quality of life [37]. Despite this, ferric maltol is not recommended for patients with active IBD due to the potential risk of increased inflammation in the GI tract [38].

### 2.3. Sucrosomial Iron

This is a newer generation of oral iron that has been associated with high intestinal absorption and bioavailability. It is a preparation of ferric pyrophosphate transported within a phospholipid and sucrose esters of fatty acid membrane. Due to its lack of direct contact with the intestinal mucosa, there has been a low report of side effects [39]. It also requires a lower drug dose than the ferrous variants [40]. Adverse events reported with this drug include GI upset nausea and vomiting, a change in bowel habits, tenesmus, abdominal pain and intestinal bleeding. However, all events were reported as mild and none were directly associated with the medication itself [40]. There are limited studies with this drug in IBD patients, however, initial clinical data suggest that this may be a viable alternative [31].

### 2.4. Advantages and Disadvantages of Oral Iron

The advantages of oral iron include a well-established safety profile, its ease of route of administration and reduced cost [3]. In addition, oral iron is not associated with reduced productivity [41]. However, due to a high proportion (90%) of non-absorbed iron remaining in the gut, there are a considerable amount of side effects with oral iron replacement, which can lead to discontinuation of treatment in 20% of patients [23,30,31]. Nausea, vomiting, altered bowel habits and abdominal pain are the most commonly experienced side effects. Nausea and abdominal discomfort are dose related and can occur 1–2 h after ingestion [42]. In this instance, the patient can trial delayed release enteric coated iron tablets. However, these tablets have reduced efficacy as they dissolve slowly in the duodenum where the majority of iron is absorbed [10,24]. In an attempt to improve tolerance and compliance, efforts are being made to develop newer iron formulations that have a reduced side effect profile. Data thus far suggest that newer iron formulations may be superior to traditional ferrous iron. However, in the absence of any head-to-head trials, it is not yet possible to draw any definitive conclusions [3].

Whether therapeutic oral iron is effective with active IBD remains uncertain, perhaps due to a lack of definitive efficacy data. However, there is established evidence that the presence of iron in the gut increases IBD activity by inducing oxidative stress at the site of bowel inflammation. Ferrous compounds are oxidated within the gut lumen or mucosa, which results in the release of activated hydroxyl radicals and the generation of reactive oxygen species [4]. These molecules can subsequently damage the intestine, causing a range of gastrointestinal symptoms [4,43]. Additionally, the presence of mucosal inflammation can further exacerbate ongoing iron malabsorption. Enteric coated ferrous formulations avoid this oxidation process by minimising iron release in the stomach but by doing so, also prevent iron absorption [10].

Oral iron also has a significant impact on microbial composition, which plays a central role in the pathogenesis of IBD. Oral iron can disrupt the gut bacterial diversity leading to change in disease activity with some bacteria more dependent on iron than others [44].

## 3. IV Iron

The updated guidelines from ECCO advise that first line treatment should be IV iron in patients with active IBD, severe anaemia (Hb < 100 g/L), if previously intolerant to oral iron and for patients in need of concomitant treatment with erythropoietin (Epo) [20]. The cumulative dose for iron repletion is based on the patient’s Hb and body weight, which can be calculated using the Ganzoni formula (total iron dose = (actual body weight × (15 − actual Hb) × 2.4 + iron stores) [45]. However, this is not consistently used in clinical practice as it is error-prone, inconvenient and underestimates iron requirements [46]. Additionally, specific product labels state specific dosing regimens which is oft preferred [45]. A simplified fixed-dose regimen for ferric carboxymaltose has been found to be more superior than the Ganzoni formula and is based on Hb and body weight. This formula can be applied in other iron formulations, which are also given at doses of 1000 mg [46].

### 3.1. Parenteral Iron Preparations

The selection of the IV agent is dependent on multiple factors including patient preference, costs and local availability [17]. It is important to note that each IV formulation follows a different protocol and, as of yet, one cannot compare the efficiency and safety profiles between the various formulations due to the lack of any large competitive trials. See Table 2 for a breakdown of the characteristics of the different IV preparations.

### 3.2. Dextran

The high molecular weight (HMW) iron replacement therapy is also known as DexFerrum [44]. It has a molecular weight of 100–500 kDa and is considered to be a stable parenteral iron product with a plasma half-life of 3–4 days [10]. The increased stability of this compound allows for high dose single administrations [1]. However, this compound has an increased risk for anaphylaxis [47] and due to its unfavourable safety profile, HMW dextran should be avoided [48]. The low molecular weight (LMW) dextran (CosmoFer), 73 kDa, can be safely administered with a 2 to 12-fold lower risk of life and non-life-threatening side effects compared to HMW dextran. Compared to the HMW dextran, there is a much smaller risk of an immunoglobulin-E mediated anaphylactoid reaction. However, this risk is still greater than the newer iron salt formulations. Only the LMW dextran is currently available for use in Europe at a maximum single dose of 200 mg administered over a minimum of 30 min. [49]

### 3.3. Sucrose (Venofer, Vifor)

Iron sucrose has a molecular weight of 34–60 kDa and is released quickly into circulation due to its relatively fast half-life of 5–6 h [10,50,51]. It is the most extensively studied IV iron preparation in IBD patients [52]. A slow injection of 100–200 mg (300 mg in some countries) given 2–3 times a week over 30 min is advised. It can also be infused slowly at a maximum dose of 500 mg once a week [50]. If the infusion is given too quickly (>4 mg/min) or a single total iron dose is too high (>7 mg/kg), the non-transferrin bound iron may cause dyspnoea, tachycardia and transient hypotension [53]. Iron sucrose is safe and well-tolerated even if previously intolerant to other IV iron products and does not carry the risk of dextran-induced anaphylactic reactions. Studies have shown an association between an improvement in QoL and treatment response rate (65–75%), which should be noticeable within 4–8 weeks. Furthermore, this preparation can be used after the first trimester in pregnancy and post-partum [10]. The main disadvantage to iron sucrose is that multiple sessions are needed in severe anaemia to replace the estimated iron deficit [50].

### 3.4. Ferric Carboxymaltose (Ferrinject)

Alongside iron sucrose, this is also extensively studied and used in IBD patients [17]. It is highly stable and has a molecular weight of 150 kDa [24,54]. It can be safely administered at a dose of 1000 mg within 15 min [23]. Multiple studies have confirmed the superiority of ferric carboxymaltose in IBD patients compared to oral iron, demonstrating the drug to be efficacious and well-tolerated with a faster rise in Hb [55]. Compared to iron sucrose, it proved to have greater efficacy and compliance with a good safety profile [56]. A potential consequence seen with carboxymaltose is hypophosphataemia. This is usually asymptomatic and evident within 24 h of administration, reaching a peak in 1 to 2 weeks [57]. Phosphate concentrations generally return to baseline by six weeks. If patients receive multiple or recurrent infusions, hypophosphataemia may last for longer periods and can rarely result in osteomalacia [57].

### 3.5. Ferric Gluconate (Ferrlecit, Sanofi)

This is a labile iron formulation, so iron is released quickly [58]. It has a molecular weight of 37 kDa [24]. This preparation is not typically used in IBD patients with more extensive studies conducted in chronic kidney disease (CKD) patients, demonstrating adequate safety and efficacy in patients on haemodialysis. It is generally given over eight infusions of 125 mg during eight consecutive dialysis sessions [58]. Side effects include nausea, hypotension, tachycardia, dyspnoea and oedema of the lungs, hands and feet. These symptoms should not be mistaken for anaphylaxis [10].

### 3.6. Ferumoxytol

This is a new iron salt formulation with its use not yet approved in Europe [59]. It has a much larger complex (750 kDa) which allows the drug to be given rapidly in large doses [60]. The dose is 510 mg administered over less than one minute with a second dose given 3–8 days later [59]. It is currently being used for IDA in CKD with limited data in treatment of anaemia in IBD. Thus far, ferumoxytol has the highest rate for adverse events per million units sold and carries a boxed warning for possible life-threatening allergic reactions [61]. Additionally, ferumoxytol can interfere with MRI examinations due to its paramagnetic nature [62].

### 3.7. Ferric Isomaltoside (Monofer)

This is the latest iron salt formulation and is found to be stable with little labile iron and a very small immunogenic potential [23,63]. The dose is 20 mg/kg, administered within 15 min [24,64]. Advantages include: full iron repletion being achieved in a single infusion, well-tolerated, safe and effective with no reports of anaphylactic or delayed allergic reactions [23]. Stein et al. demonstrated efficacy with ferric isomaltoside in IBD patients with a rise in Hb levels and a reduction in faecal calprotectin levels. The study also demonstrated a good safety profile with only 8% of patients reporting adverse drug reactions, the most common being gastrointestinal upset and skin reactions. However, this was a non-interventional study with a small sample size of only 197 patients [65].

### 3.8. Advantages versus Disadvantages of IV Iron

The primary advantage of IV iron is that it bypasses absorption through the gastrointestinal tract, thereby avoiding further mucosal aggravation and inflammation and producing less side effects. Clinicians also do not have to worry about patient’s adherence to medication [47]. Moreover, IV iron can replace iron stores and improve anaemia quicker and more efficiently than oral iron [66], although the clinical significance of this is uncertain [47]. There was a meta-analysis conducted that compared IBD patients who were given IV and oral iron. The results demonstrated that IV iron was more effective at increasing Hb levels and, due to the reduced adverse events and greater tolerability, there was a lower rate of treatment discontinuation [66]. However, studies have not shown any significant differences in quality of life or disease activity in those with IV versus oral iron [47].

The common disadvantages include a low bioavailability of IV iron and the inconvenience of IV administration. Parenteral iron is more expensive and is an additional cost to the healthcare system, which is a limiting factor in its widespread use [1,47]. However, further studies have shown that the long-term costs of ineffective oral iron therapy are outweighed by the immediate costs of IV iron [67]. Additionally, severe anaphylactoid reactions to HMW dextrans have previously been reported but are now increasingly rare with the newer LMW formulations [1]. In particular, anaphylactic reactions have been reported at 0.24/1000 infusions in IBD [66] with death being reported at 1 for every 5 million doses [68].

The side effect profiles are different for each preparation, however, the most frequently reported in LMW complexes include skin flushing, itching, dyspnoea, wheezing and stridor, myalgia, hypotension, tachycardia, nausea and diarrhoea and periorbital oedema. Serious side effects such as cardiac arrest are rare and are more commonly seen with the older, dextran-containing preparations [24].

### 3.9. When to Use IV versus Oral

This is an ongoing topic of debate between clinicians with no study having provided clear, globally accepted evidence [3]. Some authors promote the use of IV iron only in IBD patients [43], while other authors acknowledge that both routes can be efficacious and safe to use, depending on the dose and formulation used [36,69]. Although oral iron remains the first line treatment for IBD patients in the United States, IV iron is now considered standard treatment for IBD patients in Europe [70]. Regardless, several studies showed that oral iron was still the preferred modality by clinicians [71]. Furthermore, despite the fact that IV iron has been proven to be clinically effective with a well-tolerated safety profile, many clinicians are still reluctant to administer IV iron for concern of its hypersensitivity reactions [50]. Table 3 summarises these findings.

## 4. Blood Transfusions

Blood transfusions are generally not recommended in IDA due to the high risk of complications [24]. Studies have shown that the generous use of blood transfusions in upper gastrointestinal bleeds has been associated with an increased mortality risk in patients as well as an increased risk of nosocomial infection rate [72]. If the decision is to proceed with a blood transfusion, it should be done with caution and should not be intended for repeated transfusions. Specifically, blood transfusions should be reserved for patients who are severely anaemic (Hb < 70 g/L), haemodynamically unstable or have co-morbid conditions such as coronary heart disease or chronic pulmonary disease [20]. Importantly, transfusions should be used only as a temporary measure to increase Hb levels and iron replacement is still needed to replenish iron stores [1].

## 5. Erythropoietin-Stimulating Agents

Erythropoietin (Epo) should be considered as second line treatment for IBD patients with severe or symptomatic anaemia refractory to IV iron. In this instance, patients may have underlying ACD which is mistaken for IDA [20]. Currently, large long-term studies on the use of Epo in IBD is lacking with more data for patients with CKD [20,73]. There are several disadvantages to using Epo, including risk of thromboembolic events, stroke, cardiovascular events and death [74]. This is an important factor as IBD already has a high risk of venous thromboembolic events, particularly in high-risk patients with ulcerative colitis and in active disease [23]. The target Hb should be 130–140 g/L and a rapid increase of greater than 100 g/L in 2 weeks has been shown to be harmful for patients with CKD. ECCO recommends a target Hb of less than 120 g/L to minimise side effects, which include hypertension, oedema, fever, dizziness and rare red cell aplasia secondary to anti-Epo antibodies. Giving Epo can cause an increased demand for iron in the bone marrow, thus IV iron should also be given, aiming for a target ferritin level greater than 200 ug/L to prevent functional iron deficiency [20].

## 6. Follow Up and Maintenance

The goal of iron supplementation, whether given IV or orally, should be the complete normalisation of Hb and iron storage. An adequate response is an increase in Hb of at least 200 g/L or normalisation within 4 weeks of treatment. However, depending on the severity and underlying cause of anaemia, Hb correction can take up to three months or longer to replenish iron stores. Iron storage can be assessed with ferritin and transferrin saturation. This should be checked 8–12 weeks after the last IV treatment as levels can be falsely elevated [1]. When using oral therapy, ferritin can increase within 1–2 weeks. Treatment response can also be demonstrated more rapidly with reticulocyte count and reticulocyte production index, which is increased within 1–2 weeks of oral or IV iron. When giving IV iron, ferritin levels of 800 ug/L and transferrin saturations of 50% should be used as the upper limits to prevent iron overload [24]. When deciding to stop iron therapy, iron storage levels should be above the lower limits of normal to prevent risk of recurrence [20].

Unfortunately, iron deficiency and IDA can recur often in IBD patients with ongoing risk factors. Post iron replacement, ferritin levels of <100 ug/L are more likely to recur within the first four months, 100–400 ug/L within the next year and >400 ug/L after two years [75]. Faster recurrence correlates with disease activity even if patients are asymptomatic and found to have low inflammatory markers [24]. Thus, appropriate IBD management should be undertaken concomitantly with iron replacement to prevent further iron losses [1]. Following correction of iron, IBD patients should be monitored for IDA recurrence every 3 months for at least one year and then 6–12 months thereafter. This includes checking Hb, ferritin, transferrin saturations and CRP. Vitamin B12 and folic acid should be measured on a yearly basis for patients and more frequently in patients with extensive small bowel resection particularly if there is ileal involvement including ileal pouch surgery. Once ferritin falls below 100 or Hb drops below normal levels, treatment should be restarted using the same route of administration as the initial formulation [20].

## 7. Conclusions

IDA is the most common extraintestinal manifestation of IBD, with debilitating consequences on patients’ QoL. Treatment should be initiated as soon as IDA has been identified and, once corrected, levels should be monitored routinely to avoid recurrence. There is no consensus amongst gastroenterologists as to which iron preparation is best. Oral iron formulations are restricted in their use due to poor tolerability and patient compliance issues, although there are promising data for the newer oral iron formulations. Although IV iron formulations have been shown to be better tolerated and lead to a faster Hb rise than oral iron, there is still hesitancy amongst gastroenterologists to promote this administration due to its hypersensitivity risk. Large comparative trials to determine safety and efficacy between different formulations are now needed to establish clinician unanimity for standard treatment guidelines.

## Figures and Tables

**Figure 1 nutrients-12-03478-f001:**
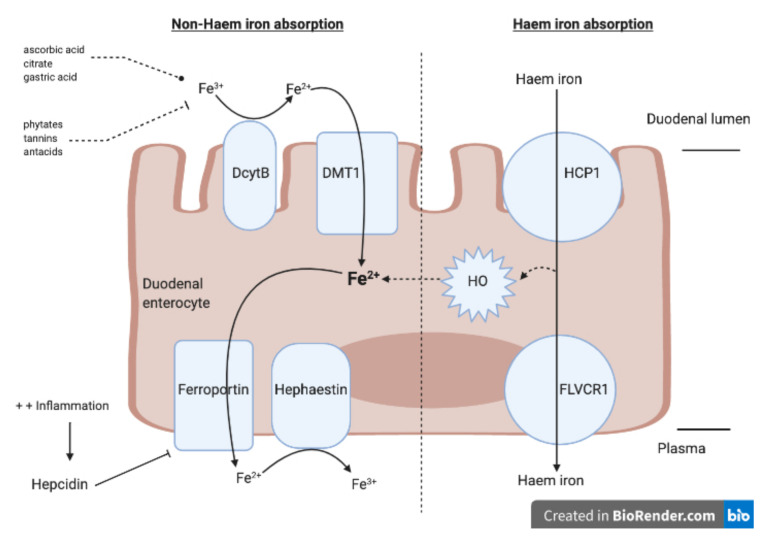
Absorption of haem and non-haem iron through the duodenal enterocyte. Non-haem absorption pathway: Ferric iron (Fe^3+^) is first reduced to ferrous iron (Fe^2+^) by the enzyme, Duodenal cytochrome B (DcytB). Fe^2+^ can then be absorbed across the apical surface via the divalent metal transporter 1 (DMT1) and exported via ferroportin. Haem absorption pathway: Haem iron is absorbed directly into the enterocyte through haem carrier protein (HCP1). Once inside the enterocyte, haem iron can either be released into plasma via the FLVCR1 receptor or be converted to Fe^2+^ through the enzyme haem oxidase (HO). Before exportation, Fe^2+^ is oxidized back to its Fe^3+^ form via Hephaestin. Hepcidin controls ferroportin and will inhibit iron export in the presence of inflammation. Facilitators of iron absorption include ascorbic acid, citrate and amino acids while inhibitors include phytate, tannins and antacids.

**Table 1 nutrients-12-03478-t001:** Diagnosing and managing anaemia in inflammatory bowel disease (IBD).

Type of Anaemia	Definition of Anaemia	Diagnosis	Microscopic Findings	Management
IDA	Women: Hb < 120 g/L Men: Hb < 130 g/L Pregnancy: Hb < 110 g/L	Low serum iron Low ferritin (<30 ug/L) Serum ferritin < 100 ug/L in inflammatory disease Low transferrin saturation (TSAT < 20%) Transferrin levels increased Reduced MCH	Microcytic and hypochromic erythrocytes	Oral or IV replacement Blood transfusion
Vitamin B12 deficiency		Low B12 levels Elevated methylmalonic acid Elevated total homocysteine Elevated MCH	Megaloblastic anaemia	High dose IV or oral B12 replacement
ACD		Low reticulocyte count Low iron Low TSAT (<20%) Transferrin levels normal or decreased Normal or raised Ferritin (<100) Low or normal MCH	Normochromic and normocytic erythrocytes	Treatment of underlying condition Blood transfusions Erythropoiesis stimulating agents
IDA and ACD		Normal or raised transferrin saturation Normal or reduced MCH	Hypochromic erythrocytes	

IDA: iron deficiency anaemia, ACD: anaemia of chronic disease, MCH: mean corpuscular haemoglobin, Hb: haemoglobin.

**Table 2 nutrients-12-03478-t002:** Characteristics of different intravenous iron preparations.

	Molecular Weight	Half-Life	Administration	Disadvantages/Risks
HMW Dextran (DexFerrum)	100–500 kDa	3–4 days	Single dose	Anaphylactoid reaction
LMW Dextran (CosmoFer)	73 kDa	5–20 h	Maximum single infusion of 20 mg/kg over 4–6 h	Immunoglobulin-E mediated anaphylactoid reaction
Sucrose (Venofer)	34–60 kDa	5–6 h	Single infusion up to 200 mg over 30 min	Multiple sessions needed in severe anaemia
Carboxymaltose (Ferrinject)	150 kDa	7–12 h	Single dose infusion of 1000 mg over 15 min (max dose of 20 mg/kg)	Hypophosphataemia
Isomaltoside (Monofer)	1000 kDa	1–4 days		Limited data in IBD
Gluconate (Ferrlecit)	37 kDa	1 h	8 infusions of 125 mg	Not currently for use in IBD
Ferumoxytol	721 kDa	14–21 h	510 mg can be given in less than 1 min	High rate of adverse events Interference with MRI

HMW: high molecular weight; LMW: low molecular weight; IBD: inflammatory bowel disease; MRI: magnetic resonance imaging.

**Table 3 nutrients-12-03478-t003:** Advantages and disadvantages between oral and intravenous iron.

	Oral Iron	IV Iron
Choice of administration	Mild IDA (Hb > 100 g/L) Quiescent IBD	Severe anaemia (Hb < 100 g/L) Intolerant to oral iron Moderate to severe IBD activity
Pros	Greater availability Ease of administration Low cost	Bypasses GI tract absorption Less side effects
Cons	Side effects with poor patient tolerance Discontinuation in 20% Increases IBD activity Disrupts microbiome	Low bioavailability Inconvenience of IV application Greater cost Risk of hypersensitivity reaction *

* Hypersensitivity risk rare in LMW (low molecular weight) dextran and newer IV formulations. IDA: iron deficiency anaemia; GI: gastrointestinal; Hb: haemoglobin; IBD: inflammatory bowel disease.

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
