# Peer review of "Iron Therapy in Inflammatory Bowel Disease"

_nutrients, 2020, doi:10.3390/nu12113478_

Round 1

Reviewer 1 Report

The paper is well written and deals with a topic that maintains a great interest among those treating patients with IBD. There are few minor points to be addressed:

Page 4, lane 127: "50-200mg/day once daily" is redundant, skip "/day"

Page 9, lane 321: "with ferritin and transferrin saturations"; saturation, not saturations.

Some formula for calculation of the amount of iron that needs to be given in order to correct anaemia and replenish iron stores by means of iv iron supplementation should be provided.

Author Response

Dear Reviewer,

Many thanks for your insights and comments into the paper. I have made the minor changes suggested on page 4 and 9. In the IV iron section, I have also included a few sentences on the Ganzoni formula as well as the newer simplified method used for ferric carboxymaltose- this has been highlighted in the text with references adjusted appropriately.

Kind regards

Aditi

Reviewer 2 Report

This is  a written overview of IDA and its management in patients with IBD.

In the section on oral iron, every other day prescription should be added addressing the initial increase in Hepcidin after oral iron ingestion.

Author Response

Dear Reviewer,

Many thanks for your comments and insights into this review paper. As per your suggestion, I have added in information regarding the benefits of every other day prescription for oral iron, explaining about the transient rise in hepcidin. I have highlighted these changes in the manuscript.

Kind regards

Aditi

Reviewer 3 Report

I found this manuscript to be overall well informed, comprehensive, accurate, and well written for its intended audience, but I have a number of comments for the authors’ consideration which I think that could improve the final version of the manuscript:

-In general, iron metabolism description must be improved: iron metabolism should be described in detail and the role of DcytB and Hephaestin should be included in this heading,taking into account the key role of these proteins.

- The role of heme receptor and heme iron absorption should also be discussed in this paragraph.

- Figure 1 represents Fe absorption through the duodenal enterocyte, however the role of DcytB in the apical membrane and Hephaestin in the basolateral membrane are not featured and they are quite relevant, having a key role in iron metabolism. They should be featured in the figure, therefore this must be improved.

-Lines 97-98: “There is also an increased risk of infection and alterations of thyroid hormones, catecholamines and neurotransmitters”. The clinical implications on basal metabolism and cognitive impairment (depression) should be discussed in detail here.

Author Response

Dear Reviewer,

Many thanks for your comments and insights into this review paper. As per your suggestions, I have added in further information in the iron metabolism section of the paper. I have also adjusted Figure 1 to incorporate the new information included. 

I have also discussed the clinical implications on basal metabolism and cognitive impairment in the clinical manifestations of IDA section of the paper.

Due to the word count, I have tried my best to provide accurate but concise information in the manuscript. 

Kind regards

Aditi